# N-Functionalization of 5-Aminotetrazoles: Balancing Energetic Performance and Molecular Stability by Introducing ADNP

**DOI:** 10.3390/ijms232415841

**Published:** 2022-12-13

**Authors:** Jin Xiong, Jinjie Chang, Jinxiong Cai, Ping Yin, Siping Pang

**Affiliations:** 1School of Materials Science and Engineering, Beijing Institute of Technology, Beijing 100081, China; 2Beijing Institute of Technology Chongqing Innovation Center, Chongqing 401120, China

**Keywords:** 5-aminotetrazole, N-bridging functionalization, nitropyrazole functionalization, high-nitrogen tetrazole

## Abstract

5-aminotetrazole is one of the most marked high-nitrogen tetrazole compounds. However, the structural modification of 5-aminotetrazole with nitro groups often leads to dramatically decreased molecular stability, while the N-bridging functionalization does not efficiently improve the density and performance. In this paper, we report on a straightforward approach for improving the density of 5-aminotetrazole by introducing 4-amino-3,5-dinitropyrazole. The following experimental and calculated properties show that nitropyrazole functionalization competes well with energetic performance and mechanic sensitivity. All compounds were thoroughly characterized using IR and NMR spectroscopy, elemental analysis, and differential scanning calorimetry (DSC). Two energetic compounds (**DMPT-1** and **DMPT-2**) were further confirmed by implementing single-crystal X-ray diffraction studies. Compound **DMPT-1** featured a high crystal density of 1.806 g cm^−3^, excellent detonation velocity (*v_D_* = 8610 m s^−1^), detonation pressure (*P* = 30.2 GPa), and impact sensitivity of 30 J.

## 1. Introduction

High energy density materials (HEDMs) remain a significant class of materials chemistry because energetic material has been widely applied in the military and civilian fields [1]. However, with the development of science and technology, higher requirements are put forward for new energetic materials, including high energy levels, low sensitivities, excellent thermal stability, and facile preparation. High positive heats of formation and densities constitute two significant parameters for the energetic properties of energetic compounds, where detonation pressures and velocities are proportional to the square of the densities [2]. Thus, developing effective strategies for increasing the densities of energetic materials is highly desirable.

In recent years, nitrogen-rich heterocycles have emerged as a new class of high energy density materials (HEDMs), which have been developed to meet the needs of national defense and environmental protection. Owing to their high thermal stability, high heat of formation, and enormous ring tension of azoles with C-N and N-N bonds, they have been the main building blocks for the design and preparation of nitrogen-rich materials [3,4,5,6]. Compared with other azole units, tetrazole is a highly stable heterocyclic structure with an extremely high nitrogen content and heat of formation, which may endow tetrazole-based energetic compounds with high performance and environmentally friendly properties [7,8,9,10,11].

Furthermore, 5-aminotetrazole (**AT**) provides a platform for a variety of functionalized tetrazoles, which have broad applications in almost all areas of energetic materials [12]. The functionalization of **AT** was mainly focused on modifying the tetrazolyl backbone and the substituted group, respectively. As can be seen in Figure 1, the introduction of explosophores, e.g., azido, nitro, and nitramino groups, enhances the energetic performance remarkably. Nevertheless, balancing high energy with stability is still a highly challenging task. Based on the literature, very few outstanding compounds possess superior overall performance compared to the benchmark energetic materials [13,14,15]. For most high-energy derivatives, further practical applications are impeded by relatively poor thermal stability and sensitivity.

Inspired by the attractive chemistry of high-nitrogen materials, a synthetic method for efficiently preparing 1-substituted 5-aminotetrazoles from cyanogen azide and amine was developed (Figure 2a) [16]. The excellent reaction scope enables this approach to access various monocyclic and bicyclic **AT** derivatives [17]. In 2016, a new approach for various pyrazole derivatives with 2-haloethylamines, followed by a reaction with cyanogen azide, resulted in ethylene-bridged 5-aminotetrazole and nitropyrazole (Figure 2b) [18]. Most of these compounds have good thermal stability and high heat of formation, however, the density of aminotetrazole-based derivatives could meet the standard of high-density compounds. As shown in Figure 2d, most **AT**-based compounds display a density below 1.70 g cm^−3^.

Five-membered azoles are commonly used as the framework for the construction of nitrogen-rich heterocyclic energetic materials, such as imidazole, pyrazole, triazole, and tetrazole [19,20,21,22]. Pyrazole energetic compounds have good thermal stabilities and low sensitivities, which have been considered a beneficial building block for energetic materials [23,24]. For example, 4-amino-3,5-dinitropyrazole (**ADNP**) has been widely used in developing energetic materials due to their good density, low friction and impact sensitivities [25,26,27]. Herein, we reported our latest progress on the energetic derivatives of **AT**. By incorporating **ADNP**, the bicyclic **AT**-based compounds **DMPT-1** and **DMPT-2** are successfully accessed. Compared to previously reported **AT** derivatives, these compounds exhibit higher densities and detonation properties while retaining thermal stability and sensitivity.

## 2. Results and Discussion

### 2.1. Synthesis

The synthesis routes of compounds **DMPT** are shown in Figure 3. Compound 1-(chloromethyl)-3,5-dinitro-1*H*-pyrazol-4-amine (**CDPA**) was prepared according to similar procedures in the literature [28]. The reaction of intermediate **CDPA**, 5-aminotetrazoles, KOH and KI at 80 °C in acetonitrile for 12 h, subjected to silica gel column chromatography, gave rise to two methylene-bridged isomers 1-((4-amino-3,5-dinitro-1*H*-pyrazol-1-yl)methyl)-1*H*-tetrazol-5-amine (**DMPT-1**) and 2-((4-amino-3,5-dinitro-1*H*-pyrazol-1-yl)methyl)-2*H*-tetrazol-5-amine (**DMPT-2**). While increasing the loading of 5-aminotetrazoles enabled an increased yield of **DMPT-2**, the yield of **DMPT-1** was significantly reduced. We then increased the loading of 5-aminotetrazole and lowered the reaction temperature to 60 °C; this led to the production of **DMPT-1** with a higher yield. Compounds **DMPT-1** and **DMPT-2** were all fully characterized using infrared (IR) spectroscopy, elemental analysis, and nuclear magnetic resonance (^1^H and ^13^C NMR) spectroscopy (see Appendix A). In addition to the straightforward synthetic route, the preparation conditions and product purification are mild and simple, which contributes to the potential practical application.

### 2.2. Single-Crystal X-ray Analysis

The suitable crystals of compounds **DMPT-1** and **DMPT-2** were obtained by slow evaporation from a mixture of ethyl acetate and petroleum ether. Compound **DMPT-1** crystallizes in the monoclinic space group *P21/c* with a crystal density of 1.806 g cm^−3^ at 296 K (Z = 4). The crystal structure of compound **DMPT-1** is given in Figure 1a. The C-NO_2_ groups with C-N bonds lengths are 1.394(3) Å (C12-N2), 1.434(3) Å (C7-N1); compared to the other C-N bonds of *N*,*N*’-methylene bridge C4-N6, 1.441(3) Å and C4-N5, 1.452(2) Å, the C-N bonds lengths are shorter. Crystal packing of **DMPT-1** exhibits a face-to-face stacking with a packing index of 74.8%. The distance between the pyrazoles (purple part) and pyrazoles (purple part) faces is 2.991 Å. The distance between the tetrazoles (blue part) and tetrazoles (blue part) faces is 3.145 Å. The N-methylene-N bridge dihedral angle between tetrazoles (blue part) and pyrazoles (purple part) ring is 87.47° for compound **DMPT-1**.

Compound **DMPT-2** belongs to the monoclinic *P2_1_/n* space group with a crystal density of 1.770 g cm^−3^ at 296 K. The crystal structure of compound **DMPT-2** is given in Figure 2d. The C-N bonds lengths of C-NO_2_ groups are 1.400(2) Å (C4-N10), 1.430(2) Å (C2-N8), which are shorter than the other C-N bonds of *N*,*N*’-methylene bridge C1-N6, 1.442(3) Å, and C1-N3, 1.455(3) Å. Crystal packing of **DMPT-2** exhibits a mixing stacking with a packing index of 73.4%. The dihedral angle of methylene bridge tetrazoles (blue part) and pyrazoles (purple part) ring is 86.70° for compound **DMPT-2**.

### 2.3. Results and Discussion

The thermostability of energetic materials is one of the most significant parameters in assessing their safety level under severe environments. The decomposition temperature was determined using differential scanning calorimetric (DSC) measurements at a heating rate of 10 °C min^−1^ under a dry nitrogen atmosphere. Compound **DMPT-1** exhibits the lowest decomposition temperature of 191 °C, whereas **DMPT-2** has a higher thermal stability of 206 °C. In general, for polynitro energetic compounds, the C-NO_2_ bonds are always regarded as the weakest bonds, which can trigger the decomposition of the compound [29]. The Mayer bond orders of C-NO_2_ bonds for **DMPT-1** and **DMPT-2** were also calculated to evaluate their molecular stabilities, as shown in Figure 2a,b. The weakest Mayer bond order of C-NO_2_ bond for compound **DMPT-1** (BN = 0.794) is shorter than that of **DMPT-2** (BN = 0.797), showing that the bond strength of C-NO_2_ bonds in **DMPT-1** is stronger than that of **DMPT-2**, which could explain in part the difference in thermal stability. Generally, aromaticity is considered an important parameter for the determination of molecular stability [30,31]. Therefore, the aromaticity of each heterocycle atmosphere could also be quantitatively described through the multicenter bond orders analysis using Multiwfn3.5 [32]. As shown in Figure 2c,d, the multicenter bond order of the tetrazole ring in **DMPT-1** (B1 = 0.5589) is shorter than that of **DMPT-2** (B1 = 0.5702), which indicates that the tetrazole ring of **DMPT-2** exhibits better aromaticity than that of **DMPT-1**, which is consistent with their thermal behaviors.

To better explain the bond strength differences between compounds **DMPT-1** and **DMPT-2**, their electrostatic potential (ESP) surfaces were analyzed. As shown in Figure 3, the red and blue regions represent the positive and negative potentials, respectively. In comparison to **DMPT-1**, the difference between the maxima and minima of ESP for the branching **DMPT-2** molecule is smaller (**DMPT-1**: +49.83, −39.23 kcal/mol; **DMPT-2**: +44.96, −30.59 kcal/mol), so that **DMPT-2** exhibits a more uniform charge distribution. The C-NO_2_ and tetrazole ring on **DMPT-2** moiety exhibit minimum potential parts, which are much lower than the C-NO_2_ and tetrazole ring on **DMPT-1**, thus indicating the distinct stability of **regioisomers** as may be related to the ESP values (**DMPT-1**, *T_d_*, 191 °C; FS, 108 N; **DMPT-2**, *T_d_*, 209 °C; FS, 192N) [33]. In principle, the lower negative ESP values and more significant electronegative regions (blue areas) on the molecular surface often give rise to lower sensitivities, which rationalize the different mechanical sensitivities.

To gain insight into understanding the relationship between weak interactions and the performance in energetic materials, the two-dimensional (2D) fingerprint spectra and Hirshfeld surfaces of **DMPT-1** and **DMPT-2** were studied using CrystalExplorer [34]. Generally, the blue and red regions on the Hirshfeld surfaces represent low and high close contact populations, respectively. As shown in Figure 4, The Hirshfeld surfaces of **DMPT-1** were near “L”-shaped structures, and the Hirshfeld surfaces of **DMPT-2** were slightly distorted. In comparison to **DMPT-1**, the reddest spots are more evenly distributed for the **DMPT-2** molecule, showing that compound **DMPT-2** can form hydrogen bonds with more surrounding molecules, which suggests that the sensitivity of compound **DMPT-2** (*FS* = 192 N) is better than that of compound **DMPT-1** (*FS* = 108 N). At the same time, the weak interactions in these molecules are given in the 2D fingerprint plots. For **DMPT-1** and **DMPT-2**, the proportion of O…H & H…O and N…H & H…N in the total weak interaction was 50.1% and 52.5%, respectively. They exhibit strong hydrogen bonding interactions, theoretically explaining the lower sensitivity (*IS* = 30 J).

### 2.4. Physicochemical and Energetic Properties

Detonation performance is a significant parameter of energetic materials for evaluating practical applications. The detonation performances of two compounds are calculated by EXPLO5 (v6.01) software. The values of detonation velocity (*v_D_*) and detonation pressure (*P*) were evaluated according to the crystal densities and calculated heats of formation. The calculated detonation velocities and detonation pressures of **DMPT-1** (*v_D_* = 8618 m s^−1^, *P* = 30.3 GPa) and **DMPT-2** (*v_D_* = 8450 m s^−1^, *P* = 28.6 GPa) can be seen in Table 1. Meanwhile, two compounds both exhibit better detonation velocity and detonation pressure than alkyl-bridged analogues (**B5**, *v_D_* = 7889 m s^−1^, *P* = 23.2 GPa, **B4**, *v_D_* = 7768 m s^−1^, *P* = 21.8 GPa). Compounds **DMPT-1** (*T_d_* = 191 °C) and **DMPT-2** (*T_d_* = 209 °C) exhibit better thermal stability compared to **ADNP** (*T_d_* = 178 °C) (Figure 5).

## 3. Materials and Methods

### 3.1. Safety Precaution

In this work, compounds **DMPT-1** and **DMPT-2** are potential energetic materials that tend to explode under certain external stimuli. Therefore, the whole experimental process should be carried out using proper safety equipment, such as safety shields, eye protection, and leather gloves.

### 3.2. General Methods

All of the reactions were carried out in the air. Ammonium 4-amino-3,5-dinitropyrazolate monohydrate [37] was prepared following procedures found in the literature. Other commercial reagents and solvents were obtained from commercial providers and used without further purification. ^1^H NMR and ^13^C NMR spectra were recorded at 25 °C on a Bruker 400 MHz and 125 MHz, respectively, and TMS as the internal standard. Chemical shifts were reported in parts per million (ppm). The onset decomposition temperature was measured using a TA Instruments DSC25 differential scanning calorimeter at a heating rate of 10 °C min^−1^ under a dry nitrogen atmosphere. Infrared spectra (IR) were obtained on a PerkinElmer Spectrum BX FT-IR instrument equipped with an ATR unit at 25 °C. Elemental analyses of C/H/N were investigated on a Thermo Scientific Flash 2000 Elemental Analyzer. A BAM fallhammer and friction tester tested impact and friction sensitivities. Densities were determined at room temperature by employing a Micromeritics AccuPyc 1340 gas pycnometer. The crystal structures were produced employing Mercury 2021.1.0 software.

### 3.3. Computational Details

The heats of formation of compounds **DMPT-1** and **DMPT-2** were performed by using the Gaussian 09 suite of programs [38,39]. Gas phase heats of formation of the title compounds were computed based on an isodesmic reaction (Appendix A). The enthalpy of reaction was carried out by combining the M062X/6-311++G** energy dif-ference for the reactions, the scaled zero-point energies (ZPE), values of thermal cor-rection (HT), and other thermal factors. The solid state heats of formation were further obtained by employing Trouton’s rule according to Equation (1) (*T* represents either melting point or decomposition temperature when no melting occurs prior to decom-position) [40] (Appendix A).
*ΔH_sub_* = 188/J mol^−1^ K^−1^ × *T*(1)

### 3.4. Synthesis of Compound 1-(Chloromethyl)-3,5-dinitro-1H-pyrazol-4-amine (CDPA)

ClCH_2_I (2 mL) was added to a 100 mL round-bottom flask equipped with a condenser and ammonium 4-amino-3,5-dinitropyrazolate monohydrate (1 mmol) in DMF (2 mL) was dropped into ClCH_2_I. Then the reaction was stirred at 50 °C for 30 min. The reaction was monitored by TLC. After the reaction was complete, the reaction mixture was allowed to cool down to rt and H_2_O (10 mL) and ether (10 mL) were added to the vessel. The resulting suspension was extracted with ether (3 × 30 mL). The organic phases were combined and dried over Na_2_SO_4_. After filtration, the solvent was removed from the filtrate under reduced pressure. The acquired residue was subjected to silica gel column chromatography (Rf = 0.33; PE:EA = 4:1) to give 1-(chloromethyl)-3,5-dinitro-1*H*-pyrazol-4-amine (50 mg, 25%). ^1^H NMR (*d*_6_-DMSO): δ 7.25 (s, 2H), 6.46 (s, 2H) ppm. ^13^C NMR (*d*_6_-DMSO): δ 142.0, 130.9, 129.9, 59.8 ppm. Elemental analysis of C_4_H_4_ClN_5_O_4_ (270.06): calcd C 21.68, H 1.82, N 28.88%; found: C 21.51, H 1.79, N 28.92%. HRMS (ESI) m/z calcd for C_4_H_3_ClN_5_O_4_ (M − H)^−^ 219.98790, found 219.98890. (Appendix A; Appendix A; Appendix A)

### 3.5. Synthesis of Compound 1-((4-Amino-3,5-dinitro-1H-pyrazol-1-yl)methyl)-1H-tetrazol-5-amine (***DMPT-1***)

1-(Chloromethyl)-3,5-dinitro-1*H*-pyrazol-4-amine **CDPA** (1 mmol), 5-aminotetrazoles (3 mmol), KOH (3 mmol), KI (1 mmol) and DMF (5 mL) were added to a 100 mL round-bottom flask equipped with a condenser. Then the reaction was stirred at 60 °C for 12 h. The reaction mixture was allowed to cool down to rt and H_2_O (15 mL) was added to the vessel. The resulting suspension was extracted with ethyl acetate (3 × 30 mL). The organic phases were combined and dried over Na_2_SO_4_. After filtration, the solvent was removed from the solution under reduced pressure. The acquired residue was subjected to silica gel column chromatography (Rf = 0.22; PE:EA = 1:1) to give 1-((4-amino-3,5-dinitro-1*H*-pyrazol-1-yl)methyl)-1*H*-tetrazol-5-amine (**DMPT-1**, 108 mg ). Yellow solid, 40% yield. T_m_ = 164 °C, *T_d_* (onset) = 191 °C. ^1^H NMR (400 MHz, *d*_6_-DMSO) δ 7.41 (s, 2H), 7.16 (s, 2H), 6.91 (s, 2H) ppm. ^13^C NMR (101 MHz, *d*_6_-DMSO) δ 156.0, 141.1, 131.2, 130.6, 60.3 ppm. IR (KBr): ṽ 3499, 3388, 2163, 2050, 1980, 1628, 1580, 1527, 1509, 1474, 1429, 1372, 1329, 1315, 1296, 1237, 1087, 1017, 965, 884, 825, 791, 760, 746, 735, 706, 674, 613, 541, 501 cm^−1^. Elemental analysis of C_5_H_6_N_10_O_4_ (270.06): calcd C 22.23, H 2.24, N 51.85%; found: C 22.36, H 2.31, N 51.72%. (Appendix A, Appendix A)

### 3.6. Synthesis of Compound 2-((4-Amino-3,5-dinitro-1H-pyrazol-1-yl)methyl)-2H-tetrazol-5-amine (***DMPT-2***)

1-(Chloromethyl)-3,5-dinitro-1*H*-pyrazol-4-amine **CDPA** (1 mmol), 5-aminotetrazoles (3 mmol), KOH (3 mmol), KI (1 mmol), and DMF (5 mL) were added to a 100 mL round-bottom flask equipped with a condenser. Then the reaction was stirred at 80 °C for 12 h. The reaction mixture was allowed to cool down to rt and H_2_O (15 mL) was added to the vessel. The resulting suspension was extracted with ethyl acetate (3 × 30 mL). The organic phases were combined and dried over Na_2_SO_4_. After filtration, the solvent was removed from the solution under reduced pressure. The acquired residue was subjected to silica gel column chromatography (Rf = 0.28; PE:EA = 4:1)to give 2-((4-amino-3,5-dinitro-1*H*-pyrazol-1-yl)methyl)-2*H*-tetrazol-5-amine (**DMPT-2**, 162 mg). Yellow solid, 60% yield. T_m_ = 205 °C, *T_d_* (onset) = 209 °C. ^1^H NMR (400 MHz, *d*_6_-DMSO) δ 7.44 (s, 2H), 7.12 (s, 2H), 6.33 (s, 2H) ppm. ^13^C NMR (101 MHz, *d*_6_-DMSO) δ 167.6, 141.9, 130.6, 130.6, 65.4 ppm. IR (KBr): 3432, 3374, 2323, 2176, 2163, 2050, 2037, 1645, 1553, 1516, 1478, 1446, 1432,1315, 1289, 1200, 1180, 1082, 1010, 995, 893, 824, 797, 746, 712, 678, 663, 537, 513, 490, 411, 403 cm^−1^. Elemental analysis of C_5_H_6_N_10_O_4_ (270.06): calcd C 22.23, H 2.24, N 51.85%; found: C 22.31, H 2.35, N 51.68%. (Appendix A, Appendix A)

## 4. Conclusions

In summary, *N*,*N*-methylene bridged 5-aminotetrazole and pyrazole were synthesized according to the methylene bridge strategy, using 5-aminotetrazole as a skeleton. A simple synthetic route, including the reaction of ammonium 4-amino-3,5-dinitropyrazolate with chloroiodomethane, followed by a reaction with 5-aminotetrazoles, was proposed to form two high-density compounds, **DMPT-1** and **DMPT-2**. The two compounds were fully characterized using IR and NMR spectroscopic data and elemental analysis. Compared to similar *N*,*N*′-ethylene-bridged asymmetric compounds, the two methylene-bridged asymmetric compounds possess higher density and enhanced detonation performance. Among them, **DMPT-1** possesses a promising overall performance (*d* = 1.806 g cm^−3^, *v_D_* = 8610 m s^−1^; *P* = 30.2 GPa). The detailed comparative properties of methylene bridged isomers indicate the pivotal role the bridging strategy plays in the energetic molecular design of novel energetic materials.

## Data Availability

Not applicable.

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
