# Peer review of "N-Functionalization of 5-Aminotetrazoles: Balancing Energetic Performance and Molecular Stability by Introducing ADNP"

_ijms, 2022, doi:10.3390/ijms232415841_

Round 1

Reviewer 1 Report

In this manuscript, the authors report the synthesis, characterization and theoretical evaluation of two new high-energy molecules with improved properties (DMPT-1 and DMPT-2). For this purpose, an approach based on the introduction of 4-amino-3,5-dinitropyrazole fragment was used to increase the density of 5-aminotetrazoles. Results of calculations performed using EXPLO5 software indicate that both compounds indeed have satisfying detonation characteristics (detonation velocities and detonation pressures). I believe that the results presented in this work could be of interest to material design scientists, especially those working in the field of high-energy materials. I recommend publication of the submitted manuscript after minor corrections:

1. Authors calculated and analyzed electrostatic potential maps of two high-energy molecules to discuss their potential impact sensitivities. It is not clear why the authors didn't comment on values of ESP in the most important regions of molecular surface like the centers of aromatic rings or regions of the C-NO2 bonds. Authors are advised to take a closer look at some of the papers co-authored by P. Politzer and J. Murray (especially from 2009. to 2016.) and to discuss this part of the study in the context of these papers (for example Propellants Explos. Pyrotech. 2016, 41, 414 – 425.)

2. Authors stated that "Generally, aromaticity is considered an important parameter for the determination of molecular stability". While this is correct, it would be helpful if the authors provided some more background related to the specific case of the influence of delocalization on the properties of high-energy nitroaromatic compounds. The following paper may be helpful: RSC Adv., 2021,11, 31933-31940. Along these lines, authors also generally commented hydrogen bonding patterns of two high-energy molecules but again missed to provide some rationale as to why hydrogen bonds are important in the context of the properties of high-energy molecules (recent papers like Phys. Chem. Chem. Phys., 2021,23, 7472-7479. could be helpful).

Author Response

  1. Authors calculated and analyzed electrostatic potential maps of two high-energy molecules to discuss their potential impact sensitivities. It is not clear why the authors didn't comment on values of ESP in the most important regions of molecular surface like the centers of aromatic rings or regions of the C-NO2 bonds.

Response: The description of ESP values has been added to demonstrate the insight into aromatic rings and C-NO2 bonds.

  1. Authors are advised to take a closer look at some of the papers co-authored by P. Politzer and J. Murray (especially from 2009. to 2016.) and to discuss this part of the study in the context of these papers (for example Propellants Explos. Pyrotech. 2016, 41, 414 - 425.)

Response: The pioneering works associated with ESP from P. Politzer and J. Murray have been cited in the revised paper. The authors appreciate the suggested references in improving the quality of this paper.

  1. Authors stated that "Generally, aromaticity is considered an important parameter for the determination of molecular stability". While this is correct, it would be helpful if the authors provided some more background related to the specific case of the influence of delocalization on the properties of high-energy nitroaromatic compounds. The following paper may be helpful: RSC Adv., 2021,11, 31933-31940. Along these lines, authors also generally commented hydrogen bonding patterns of two high-energy molecules but again missed to provide some rationale as to why hydrogen bonds are important in the context of the properties of high-energy molecules (recent papers like Phys. Chem. Chem. Phys., 2021,23, 7472-7479. could be helpful).

Response: Thanks for the valuable references. The referred literature was cited.

Author Response

  1. The name of the compound DMPT-1 and DMPT-2 in the experimental part does not correspond to the structure discussed in the course of the work.

Response: We have corrected the names according to the reviewers' suggestions.

  1. In the experimental procedure for the synthesis of bicyclic compounds, the use of 5-nitro-1,2,3-triazol-4-amine is proposed, although this compound has not been mentioned anywhere before.

Response: Thank you very much for your comment. In the experimental procedure for the synthesis of bicyclic compounds, the use of 5-aminotetrazoles is proposed, and we have changed it.

  1. The elemental analysis of the compounds does not correspond to the molecular composition given in the experimental part.

Response: Thank you very much for your comment. We have changed it according to the reviewers' suggestions.

  1. There is no high-resolution mass spectrum or elemental analysis for the CDPA compound.

Response: The high-resolution mass spectrum and elemental analysis have been added.

  1. For substances purified by column chromatography, Rf and the eluent used should be indicated.

Response: Thank you very much for your comment. Detailed information of Rf and the used eluent are added.

There are also some questions that require comments from the authors of the article:

  1. The yield of CDPA is quite low - only 25%, what happens to the rest of the starting compound during the reaction?

Response: The rest of the starting compound during the reaction gives rise to bis(3,4,5-trinitropyrazolyl)methane, confirmed by full characterization.

  1. How can one comment on the significant change in the selectivity of the reaction for obtaining isomeric DMPT when the temperature regime is changed by only 20 ℃?

Response: In part of organic reactions, the difference of 20 ℃ could tailor the regioselectivity significantly. It may be attributed to the distinct energy barriers of N1-alkylation and N2-alkylation. Additionally, the steric effect and solvent effect of N-alkylation may change the reaction selectivity as well.

  1. It is known that derivatives of tetrazol-5-amine have a sufficiently high mechanical sensitivity. At the same time, the resulting compounds are insensitive to impact. I would like to see from the authors the disclosure of some details of the investigation of sensitivity to impact and friction, namely the number of repeated experiments and the weights used.

Response: The number of repeated experiments was added.

Round 2

Reviewer 2 Report

The authors made the necessary changes to the experimental part about the name of the compound, elemental composition, and identification. Most of the comments were answered satisfactorily by the authors. However, firstly, the remark about the need to compare the properties of the obtained compounds with their analogs based on ADP was ignored.

Secondly, the methods for determining the sensitivity of compounds raise questions. Authors added a number of repeated experiments for determination of sensitivity: “A BAM fallhammer and friction tester tested impact and friction sensitivities. Each sample was tested for three times.” It is good known, that impact and friction sensitivities have probabilistic character. Their determination needs a lot of repeated experiments. For example, UN recommendation on the Transport of dangerous goods, 13.4.2 Test(a)(ii) Bam drop hammer: “The limiting impact energy, characterizing the impact sensitiveness of a substance, is defined as that lowest impact energy at which the result "explosion" is obtained from at least one out of at least six trials. The series of trials is started with a single trial at 10 J. If at this trial the result "explosion" is observed, the series is continued with trials at stepwise lower impact energies until the result "decomposition" or "no reaction" is observed. At this impact energy-level, the trial is repeated up to the total number of six if no "explosion" occurs; otherwise, the impact energy is reduced in steps until the limiting impact energy is determined. If at the impact energy level of 10 J the result "decomposition" or "no reaction" (i.e. no explosion) was observed, the test series is continued by trials at stepwise increased impact energies until for the first time the result "explosion" is obtained. Now the impact energy is lowered again until the limiting impact energy is determined.”

Three tests can’t give statistically correct value. Test’s results are nor relevant. Authors should be conduct the experiment according to accepted standards or recommendations.

Author Response

Response to Referee: 2

Comment: The authors made the necessary changes to the experimental part about the name of the compound, elemental composition, and identification. Most of the comments were answered satisfactorily by the authors. However, firstly, the remark about the need to compare the properties of the obtained compounds with their analogs based on ADP was ignored.

Response: Thank you very much for your comment. Compounds DMPT-1 (Td = 191 °C) and DMPT-2 (Td = 209 °C) exhibit better thermal stability compared to ADNP (Td = 178 °C). The density of DMPT-1 and DMPT-2 ranges between 5-amino tetrazole (1.502 g cm-3) and ADNP(1.90 g cm-3). Additional discussion was added to the manuscript.

Comment: Authors added a number of repeated experiments for determination of sensitivity: "A BAM fallhammer and friction tester tested impact and friction sensitivities. Each sample was tested for three times." It is good known, that impact and friction sensitivities have probabilistic character. Their determination needs a lot of repeated experiments. For example, UN recommendation on the Transport of dangerous goods, 13.4.2 Test(a)(ii) Bam drop hammer: "The limiting impact energy, characterizing the impact sensitiveness of a substance, is defined as that lowest impact energy at which the result "explosion" is obtained from at least one out of at least six trials. The series of trials is started with a single trial at 10 J. If at this trial the result "explosion" is observed, the series is continued with trials at stepwise lower impact energies until the result "decomposition" or "no reaction" is observed. At this impact energy-level, the trial is repeated up to the total number of six if no "explosion" occurs; otherwise, the impact energy is reduced in steps until the limiting impact energy is determined. If at the impact energy level of 10 J the result "decomposition" or "no reaction" (i.e. no explosion) was observed, the test series is continued by trials at stepwise increased impact energies until for the first time the result "explosion" is obtained. Now the impact energy is lowered again until the limiting impact energy is determined." Three tests can't give statistically correct value. Test's results are nor relevant. Authors should be conduct the experiment according to accepted standards or recommendations.

Response: Thanks for the reviewer’s valuable suggestion. We have corrected our characterization method for the impact sensitivity test. According to the standard conditions, we tested six times for each sample at 30 cm using 10kg drop hammer. One explosion was found for each sample (MPT-1 and MPT-2), respectively. However, when the height of the drop hammer decreased, no explosion could be found in six times of tests.

Round 3

Reviewer 2 Report

The initial assessment of the mechanical sensitivity of new energetic compounds is an important part of the research work. This is necessary for the safety of researchers working with new compounds. The authors satisfactorily answered all questions and conducted the experiment according to the standards. The article may be accepted for publication.